# Ornithologists’ Help to Spiders: Factors Influencing Spiders Overwintering in Bird Nesting Boxes

**DOI:** 10.3390/insects12050465

**Published:** 2021-05-18

**Authors:** Ondřej Machač, Ivan Hadrián Tuf

**Affiliations:** Department of Ecology and Environmental Sciences, Faculty of Science, Palacký University Olomouc, Šlechtitelů 27, 77900 Olomouc, Czech Republic; ivan.tuf@upol.cz

**Keywords:** Araneae, bird nests, *Anyphaena accentuata*, *Clubiona pallidula*, overwintering, artificial shelter

## Abstract

**Simple Summary:**

Nesting boxes are often used to support hole-nesting birds, but are also attractive as a shelter for many invertebrates, especially for overwintering. We studied assemblages of spiders overwintering in nesting boxes in a lowland forest and the factors influencing their abundance and activity during winter. The results show that the majority of arboreal spider species use nesting boxes to overwinter and that their abundance increases with the presence of nest material. Some spider species are also active at low temperatures in winter and can resettle emptied nesting boxes during the winter season. By hanging nesting boxes on trees, ornithologists support not only hollow nesting birds, but also overwintering spiders.

**Abstract:**

Spiders are common inhabitants of tree hollows, as well as bird nesting boxes, especially in autumn and winter. Some species of spiders use bird nesting boxes for overwintering. We investigated spider assemblages in nesting boxes and how temperature influences the abundance of overwintering spiders in nesting boxes in lowland forest in the Czech Republic. The study was conducted in the European winters of 2015–2017. In total, 3511 spider specimens belonging to 16 identified species were collected from nesting boxes over three years in late autumn and winter. Almost all species were arboreal specialists. The dominant species were *Clubiona pallidula*, *Anyphaena accentuata*, *Platnickina tincta,* and *Steatoda bipunctata*. Although the tree species had no effect on the abundance of overwintering spiders, the presence of nest material affected the abundance of spiders in the nesting boxes (preferred by *C*. *pallidula* and *P*. *tincta*). In general, spiders resettled nesting boxes during winter only sporadically, however *A*. *accentuata* reoccupied boxes continuously, and its activity was positively correlated with the outside temperature. Nesting boxes support insect-eaters all year around—birds during spring and summer and spiders during autumn and winter.

## 1. Introduction

Trees provide many specific microclimatic and structural microhabitats such as bark, trunk, cavities, hollows, and foliage microhabitats [1,2,3]. Spiders can be found in all microhabitats on trees: in spaces around the roots and trunk in soil [4], in foliage in the canopy [5], as well as on the bark [6]. Some species are specialist or temporal dwellers in tree hollows and cavities [7,8,9].

Tree cavities and hollows play an important role as a keystone component in biodiversity conservation [10,11]. Tree cavities provide breeding, feeding, and roosting habitats for many vertebrates, especially hole-nesting birds and mammals [12,13]. In hollows with nests (inhabited by birds or mammals), there is usually nest material with a rich abundance of prey and shelter for many invertebrates, including spiders. Spiders in bird and mammal nests are rarely studied, and only a few studies are known from Central Europe [14,15,16,17,18,19].

Spiders inhabiting tree hollows belong to web-builders, sit-and-wait predators, and active hunters [9]. Some species of spiders live in tree hollows throughout the year, whereas other spiders use trees only during a certain period of the year, mainly for overwintering [20,21,22,23]. Many spiders hibernate in silk sacs or stay non active if the temperature is low [24], although some species are active and hunt during winter [25]. Winter-active arboreal hunters, such as some *Philodromus* spp. or *Anyphaena accentuata*, use tree cavities as shelter in winter during daylight and hunt for prey at night [26].

Nest material is attractive for many groups of invertebrates, for example, parasites of nesting vertebrates or commensals eating organic nest detritus and/or prey on their parasites [27]. The density and distribution of tree hollows in forests depends on the tree species and the age of forest stands [28]. Due to human management, the majority of Central European forests have changed their structure from old mixed forests to rather young monocultures of single-aged plantations, where minimum hollow trees are present [29,30]. Nesting boxes are often used to support birds and small mammals, especially in younger and single-aged forests [31,32]. Only a few studies have evaluated spider communities in bird nesting boxes [33], but, to our knowledge, none have investigated spiders overwintering there.

In this study, we investigated the community of spiders overwintering in nesting boxes in lowland forests. Specifically, we studied the influence of selected factors on its distribution such as tree species, the presence of nest material, vertebrate predator exclusion by closing the entrance with a rubber bung, and the influence of temperature on the resettlement of nesting boxes during winter.

## 2. Materials and Methods

The study was performed in a floodplain forest habitat in the Království Natural Reserve near Olomouc (Czech Republic: 49°30′36.37″ N, 17°18′1.31″ E, 205 m a.s.l.). Sampling sites were located in a mature broadleaf lowland forest dominated by lindens (*Tilia* spp.), oaks (*Quercus* spp.), ash (*Fraxinus excelsior* L.), hornbeam (*Carpinus betulus* L.), alders (*Alnus* spp.), and elms (*Ulmus* spp.), with a mean annual temperature of about 12 °C. In this area, bird nesting biology has been studied intensively [33] and several sites with nesting boxes are managed. Wooden nesting boxes (290 × 220 × 180 mm, inlet 35 mm) are mounted on trees 1.5 m above the ground. The boxes are spaced about 15 m apart in parallel lines. The boxes were mounted on trees by ornithologists in the spring of 2005, and their spatial distribution does not reflect tree species. Therefore, the frequency of host trees reflects the frequency of these trees in the stand.

Spiders were collected from the nesting boxes by individual sampling using a hand aspirator. Some boxes were emptied and the nest material was heat extracted (see below). Spiders were usually identified to the species level [34]; however, some juveniles were only identified to the genus level. The majority of nonadult specimens of *Clubiona* were subadults and were bred to adults confirming their identity as *C*. *pallidula*. We used the nomenclature according to the World Spider Catalog Version 21.5 [35]. The spiders were classified into hunting strategy guilds based on Cardoso et al. [36]. All material is deposited in the first author’s collection.

Our research aims were split into three groups:
(1)The effect of tree species and bunging nesting boxes on spider abundance

In this part of the study, we used 50 wooden nesting boxes sampled in November/December of 2015, 2016, and 2017. The nesting boxes were on 28 lindens, 16 oaks, 3 ashes, and 3 alders. Half of them were closed to birds by a bung before the start of the bird nesting season (providing vertebrate predatory exclusion for spiders). The spiders were collected once a year at the turn of November/December from the same trees each year.(2)The effect of temperature on nesting box resettlement by spiders

For the second part of our study, the same 50 wooden nesting boxes were sampled in two-week intervals from January–March 2016. Temperature was measured by a datalogger (EasyLog-USB) inside and outside of three of the nesting boxes.(3)The effect of the presence of nest material on spider abundance

For this part of the study, we sampled 126 nesting boxes at the end of November 2017. The nesting boxes were on 66 lindens, 28 oaks, 21 ashes, and 11 other trees (birch, elm, and alder). Nest material from 39 nests (35 nests of *Parus major* Linnaeus, 1758; 2 *Parus caeruleus* Linnaeus, 1758; and 2 *Ficedula albicollis* (Temminck, 1815)) consisted of dry grass, moss, and mammal hair. This material was heat-extracted in Tullgren extractors in the laboratory.

For the statistical analyses, canonical correspondence models were developed in Canoco [37]. The abundances of individual species were analysed as species-dependent variables, whereas the independent environmental variables were year, tree species, temperatures (minimal, mean, and maximal, measured inside and outside the boxes), the presence of nest material, and bung. In a preliminary analysis, the length of the gradient in species variables was calculated (gradient 4.5 SD units long), and the unimodal Canonical Correspondence Analysis was subsequently chosen. There were no rare species (1–2 specimens) collected requiring exclusion from the analysis. Species data were log transformed as skewed by zero counts by the formula Y′ = Y + 1. First, we performed a global test of significance for the explanatory variables (i.e., tree species, season, bung, nest material) to avoid Type I errors resulting from multiple comparisons. We then performed forward selection to investigate the significance of particular variables and their conditional term effect. Significance was tested by Monte Carlo permutation test (499 repetitions).

Changes in the abundance of spiders that were dependent on ambient temperatures were assessed using generalized additive models. From all tested temperatures, the outside maximum temperature was chosen as it had the highest conditional effect (pseudo-F = 17.7, explaining 6.1% of the variability) in the CCA model. Changes in the abundance of individual species reflecting this temperature were analysed in GAM using Poisson’s distribution and 2.0 df temperature term smoothness.

The influence of the nest material and a bunged entrance on spider abundance were tested by Welsch two sample t-test in R version 3.6.3 [38].

## 3. Results

### 3.1. Assemblages of Spiders in Nesting Boxes

Altogether, 3511 specimens of spiders belonging to 16 identified species (11 families) were obtained from 92% of the nesting boxes (Appendix A). Some specimens were identified to the genus level (Table 1). Almost all spiders were arboreal specialists and facultative dwellers on trees. The most abundant species were as follows: *Clubiona pallidula* (47% of all collected specimens), *Anyphaena accentuata* (24%), *Steatoda bipunctata* (13%), and *Platnickina tincta* (5%). The majority of species were common forest arboreal species, only *Pseudicius encarpatus* was rare; none of the species were listed on the Red List of Czech Spiders [39].

Four species were space web hunters or orb web hunters, and three were sheet web hunters, but the most abundant guilds were non-web hunters (9 species), with one ambush hunter, two ground hunters, and seven other hunters (Table 1).

### 3.2. Effect of Tree Species and Bunging Nesting Boxes

The spider assemblages differed significantly between the studied years (CCA: from 2015 to 2017: F = 2.6, *p* = 0.002; and 2016 to 2017: F = 3.2, *p* = 0.006, respectively; Figure 1). The tree species studied revealed no significant effect on the abundance of overwintering spiders in nesting boxes (CCA: linden: F = 0.8, *p* = 0.666; oak: F = 1.00, *p* = 0.408; ash: F = 0.30, *p* = 0.97). Nesting boxes without a bung hosted significantly more spiders than the bunged boxes (CCA: F = 2, *p* = 0.026; Figure 2).

### 3.3. Effect of Temperature on the Nesting Box Resettlement by Spiders

The temperature inside and outside of the nesting boxes did not difer significantly. Temperature had a significant effect on seven species active during winter (GAM: *A*. *accentuata*, F = 42, *p* < 0.001; *C*. *pallidula*, F = 31.4, *p* < 0.001; *C*. *brevipes*, F = 8, *p* < 0.001; *P*. *tincta*, F = 16.1, *p* < 0.001; *S*. *bipunctata*, F = 7.3, *p* < 0.001; *Scotophaeus* sp., F = 3.9, *p* = 0.021; *Tetragnatha* sp., F = 3.1, *p* = 0.046). The strongest positive effect on the resettlement of the inspected boxes by spiders had the maximum temperature outside the boxes (CCA: t-max outside: F = 17.70, *p* = 0.002; t-max inside: F = 7.10, *p* = 0.002; t-mean outside: F = 4.20, *p* = 0.002; t-mean inside: F = 3.80, *p* = 0.002; t-min outside: F = 3.60, *p* = 0.002; t-min inside: F = 2.50, *p* = 0.006). There were two dominant species with opposite patterns: *C*. *pallidula* was abundant in nesting boxes at lower maximal temperatures (Figure 3), whereas *A*. *accentuata* was significantly more abundant in boxes following higher outside maximal temperatures (Pearson’s R = 0.72; Figure 3).

### 3.4. Effect of Presence of Nest Material on Spider Abundances

In 2017, material was collected from 126 nesting boxes, of which 39 boxes contained nest material (grass, moss, hair). The majority of the nests were built by the great tit (*Parus major*). A total of ten spider species were heat-extracted from the nesting material in the laboratory (Table 1). The most common species were *C*. *brevipes* (37% of the specimens collected), *C*. *pallidula* (35%), and *P*. *tincta* (26%). Other invertebrates extracted included mites, fleas, earwigs, and the larvae of Diptera and Lepidoptera. Nest material significantly supported the higher abundance of spiders in the nesting boxes (Welch two sample *t*-test, F = 8.80, *p* = 0.002, Figure 4). Three spider species heat-extracted from the nest material had a significantly higher abundance than those found in the boxes without nest material (Welch two sample t-test: *C*. *pallidula*: T = 4.14, df = 53.309, *p* < 0.001; *C*. *brevipes*: T = 2.80, df = 42.611, *p* = 0.008; *P*. *tincta*: T = 3.17, df = 41.880, *p* = 0.003). On the other hand, *A*. *accentuata* and *S*. *bipunctata* were more abundant in the empty nesting boxes (Welch two sample *t*-test: *A*. *accentuata*: T = −2.65, df = 123.970, *p* = 0.009; *S*. *bipunctata*: T = −2.61, df = 121.380, *p* = 0.010, Figure 5).

## 4. Discussion

We analysed the assemblages of spiders overwintering in nesting boxes, as well as several environmental factors affecting their abundance in nesting boxes in a lowland forest. As we know, spiders are common invertebrates in nesting boxes in different habitats not only in winter, but also during the whole year [33,40,41]. The abundance of spiders in nesting boxes are higher per nest than for other bird nest habitats, e.g., in burrow nests [18] or free nests [42]. Comparing to our results, more species of spiders were collected from bird nests in a similar study in Slovakia. However, in that study, the spiders were collected from spring to autumn (October) [19]. The highest abundance of spiders in nesting boxes was in late autumn, because spiders migrate to boxes for overwintering. During the spring and summer, spiders are preyed on by nesting birds [19], whereas the predatory impact by birds in autumn and winter is only low. Spiders were missing in nesting boxes where birds roosted. Few such boxes were recognisable by the presence of droppings.

Almost all species obtained from nesting boxes are common arboricolous species [3], typical for the local lowland forests of Central Europe [6]. Only *Tegenaria silvestris* is not a typical tree inhabitant, as it generally lives on the ground and only occasionally in the lower part of tree trunks [43]. *Mangora acalypha* is better known as an herb dweller [35]. The majority of species were bark and trunk dwellers, fewer were branch dwellers, and some were typical hollow specialists [9]. The species spectrum was similar to that found by Černecká et al. [19], with *A*. *accentuata* and *Clubiona* species being most dominant [3,35]. A relatively low abundance of *Philodromus* spp. in the nesting boxes was surprising, as this species is a common inhabitant of trees and nesting boxes [19] and overwinters under bark and in hollows [22]. Some other abundant species from nesting boxes in Slovakia [19], such as *Amaurobius fenestralis* (Ström, 1768) and *Segestria senoculata* (Linnaeus, 1758), generally live in coniferous forests [3,44], and were absent in our study, which focused on deciduous forests.

The most abundant hunter gild was other hunters, due to the dominant species *C*. *pallidula* and *A*. *accentuata*. A similar guild spectrum was found in nesting boxes in Slovakia [19]. Spiders in nesting boxes and on trees are most numerous in autumn [19,22] because of the need to find shelters in which to overwinter, often in hollows or spaces under the bark [6]. Some species overwinter in nesting boxes in the remains of the nest material or on the inner sides of nesting boxes in silk shelters (*Clubiona*), whereas some other species build webs in the spaces inside nesting boxes (*S*. *bipunctata*) or stay active and do not make web shelters (*A*. *accentuata*) (Figure A1 in Appendix B). Species classified as orb web builders do not build webs in nesting boxes and use nesting boxes only as shelter. *Steatoda bipunctata,* which is classified as a space web hunter, is a typical tree hollow dweller [9] that dwells in its web in nesting boxes probably for the whole year, as in tree hollows. On the other hand, another space web builder, *P. tincta*, lives on leaves in branches during the vegetation season and migrates to nesting boxes or tree hollows only for overwintering (we only found inactive specimens without webs on the inner walls of boxes or in the nest material). The most abundant species in nesting boxes, *C*. *pallidula*, overwintered inactive in silk sacs under the bark or in tree hollows [20,45,46].

In the U.S.A., McComb and Noble [40] reported arachnids in only 6.7% of nesting boxes, with the lowest abundances found in winter, whereas in our study, spiders were found in almost all (92%) of the nesting boxes. Closing nesting boxes with a bung had a significantly negative effect on the spider abundance in the boxes [19]. Spiders probably use artificial entrances to preferably colonize boxes, but small and flat species can use fissures under the roof or between the wooden walls for entering. During autumn or winter, predatory pressure on invertebrates in nesting boxes is probably not as high as in the spring and summer. Nevertheless, some species of birds (e.g., present *Parus major*) roost in nesting boxes during winter nights [47].

We did not find a significant difference between the studied tree species, in accordance with Černecká et al. [19]. This effect was expected in our study because the studied tree species did not recognizably differ in bark structure and habitat. Some spider species can prefer a specific tree and, consequently, abundances of different tree species can be significantly different [48], but such differences between broadleaf and coniferous trees may be more apparent [23].

Temperature had a significant effect on the probability of the resettlement of inspected and emptied nesting boxes by *A*. *accentuata*, which was the most active species during winter [26,49]. This species and some species of *Philodromus* are winter-active predators and prey even at temperatures close to 0 °C [48]. Other dominant species resettled emptied boxes only sporadically, and not during cold days. *Anyphaena accentuata* was the only spider able to regularly resettle emptied boxes without nest material during winter.

The nest material had a significant positive effect on the abundance of spiders. For example, *Salticus zebraneus* was found only in boxes with nest material. Nest material is both a shelter and a source of prey for spiders [27]. Spiders, especially juvenile *P*. *tincta* and *Clubiona* species (*C*. *pallidula* and *C*. *brevipes*), were overwintering in nest material. Juveniles of *Clubiona* were also found to be numerous in nest material in other studies [14,44]. *Clubiona* overwinters in silk sacs on the sides of nesting boxes and also among the nest material. Nest material is also used as shelter by *A*. *accentuata*; this species is known as an active winter predator [50], preying on invertebrates in the nest material. Only juvenile inactive specimens of *S. bipunctata* were found in the nest material, whereas adults built webs in the nesting boxes.

Nesting boxes are an artificial habitat and, during winter, offer shelter for a diverse spectrum of tree-dwelling spiders. We found about half of the arboreal spider species that are recorded in this region [6], and almost all that are known to overwinter on trees. By hanging nesting boxes on trees, ornithologists support not only hollow nesting birds and small mammals, but also overwintering spiders; the birds during the nesting season, and the spiders during winter. Both birds and spiders are important agents for protecting forests against pest insects.

## 5. Conclusions

During winter, nesting boxes offer shelter for a diverse spectrum of tree-dwelling spiders. We found about half of the arboreal tree species recorded in this region. The dominant overwintering species are *Clubiona pallidula*, *Anyphaena accentuata*, *Platnickina tincta,* and *Steatoda bipunctata*. The study revealed that tree species had no significant effect on the abundance of overwintering spiders in nesting boxes. The open nesting boxes with nest material significantly supported a higher abundance of spiders inside the boxes. Temperature had a significant effect on the probability for resettlement of the nesting box by *A*. *accentuata*, which was the most active species during the winter. We recommend that ornithologists leave nest material in the nesting boxes for the winter months without sealing them (the cleaning of nest boxes is better in early spring), thus helping spider species on the trees during the wintering season.

## Figures and Tables

**Figure 1 insects-12-00465-f001:**
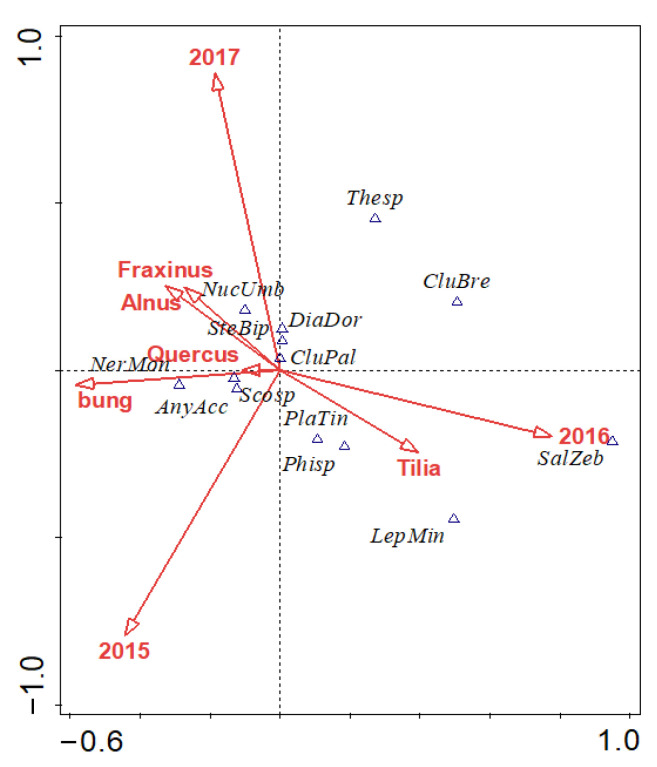
Spider species in relation to the sampling year, tree species, and bunging of nesting boxes during three years (AnyAcc–*Anyphaena accentuata*, CluPal–*Clubiona pallidula*, CluBre–*Clubiona brevipes*, DiaDor–*Diaea dorsata*, LepMin–*Lepthyphantes minutus*, NerMon–*Neriene montana*, NucUmb–*Nuctenea umbratica*, Phisp–*Philodromus* sp., PlaTin–*Platnickina tincta*, SalZeb–*Salticus zebraneus*, Scosp–*Scotophaeus* sp., SteBip–*Steatoda bipunctata*). CCA biplot is statistically significant (pseudo-F = 1.6, *p* = 0.012), and explanatory variables account for 7.0% of the variability.

**Figure 2 insects-12-00465-f002:**
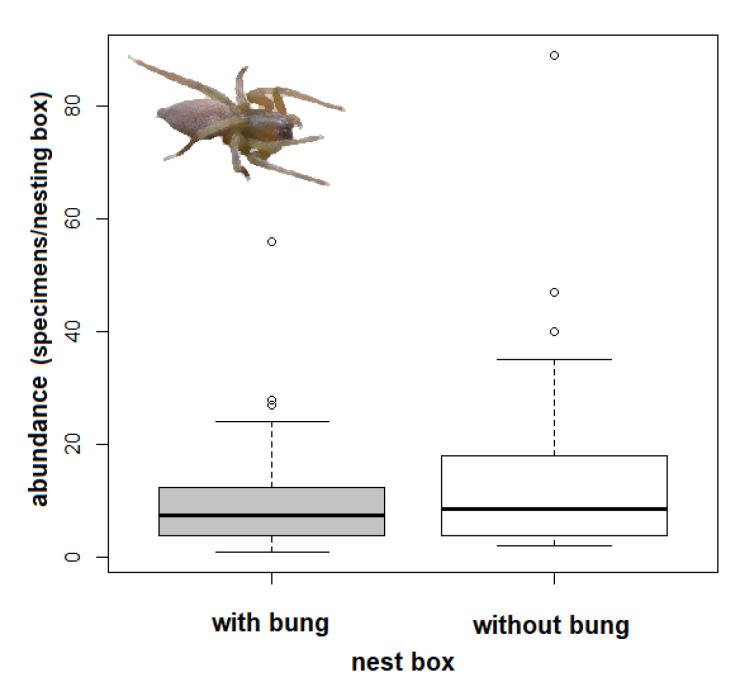
Comparison of the number of spider specimens per box and inspection (abundance = individuals) in bunged nesting boxes and boxes without a bung (sampled once in winter of 2015, 2016, and 2017), bold lines in the box plot are the median and the lines are 95% CI, the dots are outliers.

**Figure 3 insects-12-00465-f003:**
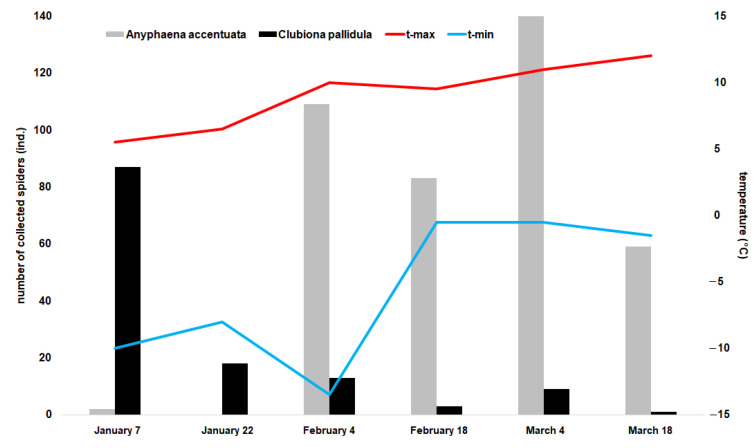
Abundance of *Anyphaena accentuata* and *Clubiona pallidula* per box during winter and the maximum and minimum temperature outside the boxes measured 14 days prior to spider collection in 2016.

**Figure 4 insects-12-00465-f004:**
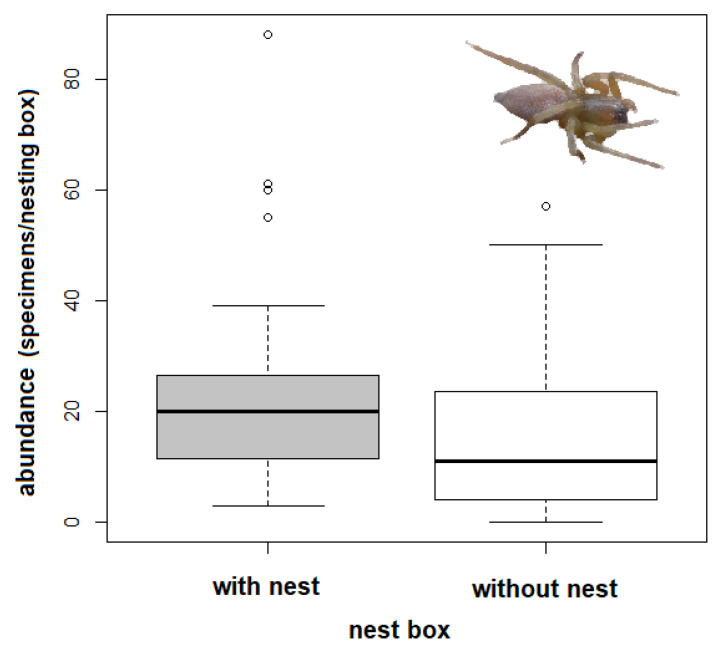
Comparison of the number of spider specimens per box and inspection (abundance = individuals) in nesting boxes with and without nest material), bold lines in the boxplot are the median and the lines are 95% CI, the dots are outliers.

**Figure 5 insects-12-00465-f005:**
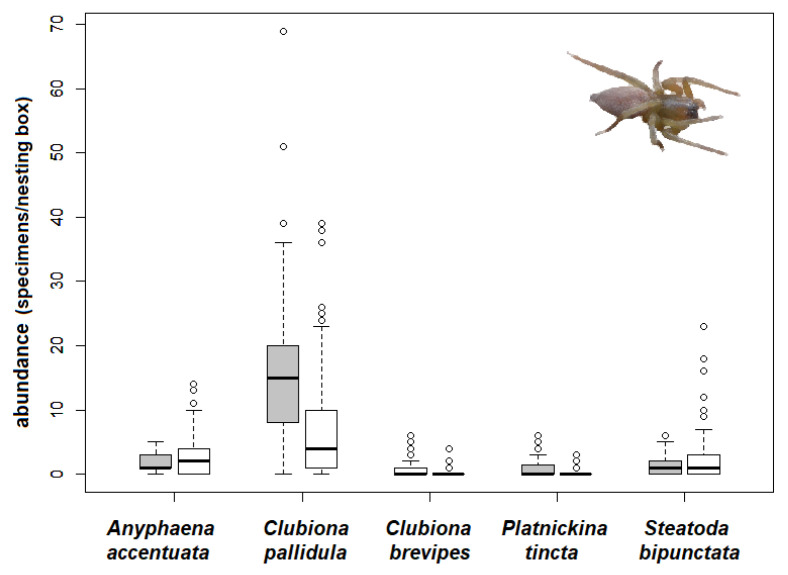
Number of spider specimens per box and inspection (abundance) of dominant spider species in nesting boxes with nest material (grey) and without nest material (white), the bold lines in the boxplot are the median and the lines are 95% CI, the dots are outliers.

**Table 1 insects-12-00465-t001:** Species list of spiders overwintering in nesting boxes and their numbers. Hand collected = number of spiders hand collected in 2015–2017, heat extracted = number of spiders heat-extracted from nest material from 39 nesting boxes in 2017.

Species/Families	Ecological Niche	Hunter Guild	Hand Collected	Heat Extracted
Agelenidae				
*Tegenaria silvestris* L. Koch, 1872	ground, trunk base	sheet web	3	0
Anyphaenidae				
*Anyphaena accentuata* (Walckenaer, 1802)	branches, bark	other hunters	851	67
Araneidae				
*Mangora acalypha* (Walckenaer, 1802)	herbs, branches	orb web	1	0
*Nuctenea umbratica* (Clerck, 1757)	bark	orb web	14	0
*Zilla diodia* (Walckenaer, 1802)	branches	orb web	1	0
Clubionidae				
*Clubiona brevipes* Blackwall, 1841	branches	other hunters	84	31
*Clubiona pallidula* (Clerck, 1757)	bark	other hunters	1635	574
Gnaphosidae				
*Micaria subopaca* Westring, 1861	bark	ground hunters	4	2
*Scotophaeus* sp.	hollows	ground hunters	112	9
Linyphiidae				
*Lepthyphantes minutus* (Blackwall, 1833)	bark	sheet web	6	1
*Neriene montana* (Clerck, 1757)	hollows, branches	sheet web	28	2
Philodromidae				
*Philodromus dispar* Walckenaer, 1826	branches	other hunters	1	0
*Philodromus* sp.	branches	other hunters	105	25
Salticidae				
*Pseudicius encarpatus* (Walckenaer, 1802)	bark	other hunters	2	0
*Salticus zebraneus* (C. L. Koch, 1837)	bark	other hunters	20	9
Theridiidae				
*Dipoena* sp.	bark	space web	2	0
*Platnickina tincta* (Walckenaer, 1802)	branches	space web	162	42
*Steatoda bipunctata* (Linnaeus, 1758)	hollow	space web	449	51
*Theridion* sp.	bark, branches	space web	14	3
Thomisidae				
*Diaea dorsata* (Fabricius, 1777)	branches	ambush hunters	12	3
Tetragnathidae				
*Tetragnatha* sp.	branches	orb web	5	0

## Data Availability

All raw data are available in Appendix A. Spiders are in the collection of the firt author.

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
