# Peer review of "Ornithologists’ Help to Spiders: Factors Influencing Spiders Overwintering in Bird Nesting Boxes"

_insects, 2021, doi:10.3390/insects12050465_

Round 1
Reviewer 1 Report
The resubmitted manuscript was clearly better. In this form, I support its publication without change. I have no further comments. Thank you for accepting my corrective comments.
Author Response
Please add your reply to reviewers

Reviewer 2 Report
This paper has improved somewhat in comparison to my initial review, but it is still let down by its poor English. I have spent an considerable amount of time cleaning this up, but ultimately gave up in the discussion, which still requires a native English speaker to improve style and clarity/contents.

Reviewer 3 Report
The authors made an effort to improve the previous manuscript. Although the English still requires language editing, in general, I am satisfied with this version and I still think this is an interesting contribution.
Some minor remarks:
Figures: see comments attached to the pdf file.
Please revise the abbreviated (genus) name of some species already mentioned before (marked in yellow) throughout the text.

Author Response
Please see the attachment.

This manuscript is a resubmission of an earlier submission. The following is a list of the peer review reports and author responses from that submission.
Round 1
Reviewer 1 Report
Good manuscript, suitable for publication. The sample size is sufficient for the examination of the topic and the issues raised. The conclusions are correct.
In my opinion, it would be worth emphasizing in the manuscript (in the Discussion) that this artificial habitat (the wooden nesting boxes and the nest material in it) is a particularly complex environment.
This complexity means that the bark, trunk, branch, foliage dwellers and some typical hollow specialist spiders at the same time occupy and colonize it.
This is also the reason for the diverse niche and guild (hunters guilds) composition of the spider fauna.
However, it can also be noted that some exclusive bark dwellers are missing or under-represented in the collection (e.g. Moebelia penicillata, Philodromus margaritatus ).
Suggestion for future work: In parallel with the collection from the nesting boxes, other collections methods should be used in order to get to know the efficiency and presumed selectivity of the method.
An important result is that the nesting boxes without bung hosted significantly more spiders than bunged boxes.
It would be nice to share exactly when the nesting boxes were closed by bung?
If the nesting boxes are sealed before the cold weather comes, there will understandably be fewer inhabitants.
I suggest referring to the publication of KUBCOVÁ L. & SCHLAGHAMERSKÝ for lines 200-202 of the manuscript. They also collected Tegenaria silvestrist from an oak trunk.
Kubcová L. & Schlaghamerský J. (2002): Zur Spinnenfauna der Stammregion stehenden Totholzes in südmährischen Auenwäldern. Arachnol. Mitt. 24: 35–61.
Suggestions for conclusions
„Tree species revealed no significant effect to abundance of overwintering spiders in nesting boxes.”
This is not necessarily generalizable, so I suggest adding “The tree species studied” to the beginning of the sentence.
The bark structure of the studied deciduous tree species was relatively similar. Trees with special bark structures, such as some coniferous trees (e.g. Pinus sylvestris), are likely to have a unique effect on the composition of spider fauna in nesting boxes.
A small addition to the statement on line 268.
The nesting boxes „with nest material” and in the open state (without bung) „significantly supported higher abundance of spiders”.
Based on the results of their studies, make a recommendation to those who handle nesting boxes.
We recommend that ornithologists leave nest material in the nesting boxes for the winter months without sealing them,, thus helping spider species on the trees during the wintering season.
Reviewer 2 Report
This manuscript appears to be an interesting contribution to forest spider ecology, but it requires a review by an native English speaker before its scientific merit can be assessed.
Reviewer 3 Report
The authors conducted a study on the effect of nesting boxes for birds on the community of arboreal spiders in a single floodplain forest. Due to loss of quantity and quality of forests, the collateral benefits of nesting boxes for non-target organisms are important. Moreover, research on this area is relevant for other landscapes such as agroecosystems where shelters for beneficial arthropods including spiders (e.g. against pests) could be extremely useful. In general, I found the manuscript of interest and relatively well written (an effort to increase the English standard should be made). My main concern is related to data analysis and its description. Although apparently the appropriate methods were applied (e.g. GAMs for count data), the modeling process is not explained; for example, how the smooth functions were developed? How the models were validated? Which distribution of error was used?
Minor remarks on the pdf version.
